# Assessing Crimp of Fibres in Random Networks with 3D Imaging

**DOI:** 10.3390/polym15041050

**Published:** 2023-02-20

**Authors:** Yasasween Hewavidana, Mehmet N. Balci, Andrew Gleadall, Behnam Pourdeyhimi, Vadim V. Silberschmidt, Emrah Demirci

**Affiliations:** 1Wolfson School of Mechanical, Electrical and Manufacturing Engineering, Loughborough University, Loughborough LE11 3TU, UK; 2Department of Mechanical Engineering, Hacettepe University, Ankara 06800, Turkey; 3The Nonwovens Institute, North Carolina State University, 1010 Main Campus Dr, Raleigh, NC 27606, USA

**Keywords:** crimped fibre, fibre crimp, non-crimped fibre, nonwovens, parametric algorithm, random fibrous network, X-ray micro-CT

## Abstract

The analysis of fibrous structures using micro-computer tomography (µCT) is becoming more important as it provides an opportunity to characterise the mechanical properties and performance of materials. This study is the first attempt to provide computations of fibre crimp for various random fibrous networks (RFNs) based on µCT data. A parametric algorithm was developed to compute fibre crimp in fibres in a virtual domain. It was successfully tested for six different X-ray µCT models of nonwoven fabrics. Computations showed that nonwoven fabrics with crimped fibres exhibited higher crimp levels than those with non-crimped fibres, as expected. However, with the increased fabric density of the non-crimped nonwovens, fibres tended to be more crimped. Additionally, the projected fibre crimp was computed for all three major 2D planes, and the obtained results were statistically analysed. Initially, the algorithm was tested for a small-size, nonwoven model containing only four fibres. The fraction of nearly straight fibres was computed for both crimped and non-crimped fabrics. The mean value of the fibre crimp demonstrated that fibre segments between intersections were almost straight. However, it was observed that there were no perfectly straight fibres in the analysed RFNs. This study is applicable to approach employing a finite-element analysis (FEA) and computational fluid dynamics (CFD) to model/analyse RFNs.

## 1. Introduction

Recent developments in microscopical imaging techniques has allowed for the detailed reconstruction of very complex fibrous structures. Before the advent of 3D imaging systems, scanning electron microscopy (SEM) provided 2D images that allowed for the identification of the relative 3D microscopical behaviour of fibrous structures. In recent years, high-resolution, X-ray µCT systems became more popular for non-destructive 3D image processing [1,2]. Such systems with sub-micron resolution capabilities (e.g., Zeiss Versa-520) are used to analyse fibres at a micro-scale level. They are employed to acquire complex, microstructural images of very different natural and man-made fibres and analyse these images with integrated software based on voxel-processing techniques.

Generally, fibrous structures can be categorised into two main groups: natural and artificial systems. For instance, fibrous structures such as nonwovens, papers, and composites are manufactured, and structures such as collagen, tendon, ligaments, and skin are examples of natural (biological) systems. Wood is another natural, organic material, filled with cellulose fibres [3]. Fibres in biological systems have been analysed at the microscopical level over many years for different scenarios. Confocal reflection microscopy was used to determine the fibre orientation and porosity of collagen networks [4,5]. A synchrotron, radiation-based X-ray diffraction system was used to develop a parametric algorithm to determine the local fibre orientation of biological tissues; this was validated using cricket samples [6]. Motaleb et al. (2021) used Fourier transform infrared (FTIR) and scanning electron microscopy to investigate the morphological structure and chemical composition of nonwoven fibres made of banana trees [7]. X-ray μCT systems are used to observe fibrous composites to evaluate damage after impact. Composites made with non-crimped, carbon-fibre-reinforced thermoplastic laminations were subjected to an impact force provided by a hemispherical impactor with a 16 mm diameter, and the resulting damage was observed using an X-ray μCT system [8].

The use of microscopy and image processing was extended to explore manufactured fibrous structures as well. Pratt and Cooper (2017) used an X-ray µCT system to determine the 3D fibre orientation of biological structures such as the vascular canals in bones [9]. For instance, the 2D fibre orientation distribution functions of fibres of random fibrous networks such as nonwovens were determined using SEM and optical microscopy [10,11]. A fibre diameter is another important parameter of a fibrous network. It can change individually along a fibre during the manufacturing process, affecting the cross-section area of a fibre. For example, 2D images obtained using optical microscopy were used to determine the cross-section area of silk and cocoon fibres. Image-processing features in Photoshop software were used to characterise the mechanical properties of the fibrous structures [12].

The geometrical fibre distribution of fibres in a network can be uniform along a specified direction or completely non-uniform in the entire fabric. Fibres in a woven fabric are slanted along a certain direction symmetrically in the entire matrix, and their orientation angle is fully flexible to change according to design requirements. However, fibre distribution in nonwovens is random, making them heterogeneous materials for several reasons, including the nature of the manufacturing process and their microstructure. Factors such as structure morphology and the geometrical properties of constituent fibres considerably affect the mechanical properties of a fibrous network. Hence, it is crucial to analyse the morphological and material properties of fibres to examine the mechanical properties of a fibrous network. The orientation distribution of fibres and fibre crimp are critical morphological parameters that significantly impact the mechanical behaviour of fibrous networks. A 3D orientation distribution function (ODF) represents the geometrical fibre orientation of the entire structure; several studies on ODFs were completed using fibrous networks and stochastic models [13,14].

Consequently, the consideration of fibre crimp for different purposes took place in various scenarios under many assumptions. For instance, Backer and Petterson (1960) predicted the tensile properties of a fibrous network without considering fibre crimp/curl by assuming that the fibre segments between bond points were straight [15]. Later, this theory was extended by adding curliness to fibre segments [16,17]. As fibre crimp/curl and fibre orientation affect the mechanical properties of an RFN, many researchers tried to incorporate both parameters into their approaches. These parameters were highly important for the stress–strain behaviour of random fibrous networks. Generally, fabric contains fibres with different levels of crimp, and each fibre experiences different levels of strain. In stretching, straight fibre segments react to the tensile force first, while slightly curved fibres sustain the strain and start to carry the load at higher levels of stretching. Highly crimped fibres can maintain their curled states until the fabric fails. It is possible to predict a stress–strain curve for any 2D case, especially along MD and CD in nonwoven fabrics. Adanur and Liao (1999) predicted the tensile properties of nonwovens based on the ODF of fibres [18]. Tensile stress in the fibre layer was estimated using a quantity of fibres within the layer and an equivalent fibre stress. To include the impact of distribution of fibre curl in all directions, the fibre modulus of the equivalent curve was replaced using the original values since this method provided an approximation of the tensile behaviour of RFNs. However, most computational methods for fibre crimp were based on experimental analysis. In recent years, a significant amount of research was implemented for RFNs using nonwovens in a non-destructive manner, while optical microscopy provides an opportunity to analyse fibrous structures in a 2D domain, it does not allow for the analysis of the distribution of fibres in the thickness direction (TD). However, high resolution X-ray µCT systems can produce data on 3D, complex random fibrous networks, even for fibre diameters at the micrometre level. In this study, six different, nonwoven fabrics were used as examples of RFNs to compute the fibre crimp in 3D. The entire experimental process was non-destructive as it was based on the X-ray µCT process and voxel-processing techniques. The algorithms developed are able to assess the fibre crimp level of fibrous structures to characterise their crimp properties. The following sections present an experimental setup using an X-ray µCT system, materials, methods, a methodology, and an analysis of the obtained results.

## 2. Materials and Methods

### 2.1. Materials

A Nikon XTH 160 Xi X-ray µCT system (Nottingham, UK) was used in this study. The developed set of algorithms was fully tested for six different X-ray µCT models of nonwovens. Selected categories included low (L)-, medium (M)-, and high (H)-density nonwovens with different combinations of manufacturing parameters (Table 1). The density of the H fabrics was 6–7 times larger than that of the L fabrics, while that of the M fabrics (a traditional, intermediate fabric density) was only approximately 3 times larger. The dry-laid fabrics studied were produced using two main manufacturing techniques called “*through-air bonding*:” (TAB) and “*thermal calendaring*” (TC) techniques; they were stacked into single or several layers. The TAB fabrics were manufactured with bi-component fibres, polypropylene (PP), and polyethylene (PE), while the TC fabrics were composed of mono-component PP fibres. All six categories contained both crimped and non-crimped fabrics. The fabrics of medium density with crimped fibres were coarse, with an average diameter approximately two times larger than that of a non-crimped fabric of the same density. Generally, coarse fibres lead to greater inter-fibre spacing, resulting in large pores in the fabric. Additionally, the surface texture of fabrics with fine fibres is smooth (compared to coarse-fibre fabrics), and the number of fibres per unit area is higher. As a result, a fabric with fine fibres appears cloudier (by visual inspection) than fabrics with coarse fibres [19,20].

### 2.2. Experimental Set-Up

A schematic of the X-ray µCT process for a nonwoven fabric is presented in Figure 1a, with a test specimen of a 3 mm * 3 mm window size shown in Figure 1b. The sample holder for tests was prepared using 500 gsm of extra-thick cardboard. The thickness of the specimens did not exceed 2 mm. One of the main challenges that the researchers experienced with the X-ray µCT system was the generation of ring artifacts in the entire image. However, in this study, the ring artifacts scattered in the image were successfully reduced by fixing the fabric onto a thick sample holder. As a result, the artifacts were moved towards the sample holder and did not affect the 3D image of the fabric. Centrelines of a test specimen and the CT holder were marked in order to align them precisely. Therefore, the specimens were always within the X-ray beams for the entire 360° rotation.

Three-dimensional models of the nonwovens were reconstructed using 2D tiff images, generated by the µCT system. The duration of the scanning time was up to 10.5 h, with eight frames produced in each projection as 2D tiff files, improving the quality of the volumetric image. A 2.5 µm resolution was used, while low-energy conditions were set at 50 kV and 50 µA during the entire period of scanning. However, fibres were exposed for around 1 s in each projection, which is a relatively high exposure time compared to X-ray µCT scanning of high-density materials such as metals. This provided sufficient time for the X-rays to clearly detect fibres with edges and avoid the generation of discontinuous fibre segments in the final reconstructed image. As automatic detection was not suitable for low-density materials such as the nonwovens, finding the centre of rotation was a challenging. However, based on acquired experience, it was located at 1 voxel length for all X-ray µCT models of this study. Generally, nonwoven fabrics are characterised by two principal directions: a machine direction (MD) and a cross direction (CD). The MD is the flow direction of the web assembly on the conveyor during the manufacturing process. The CD is perpendicular to the MD on the plane of conveyor where the web assembly forms the sheet geometry, and the third axis governs the thickness direction (TD) of the fibrous structure.

## 3. Methodology

Nonwovens are used for many different applications. They do not exhibit clear, repeatable microstructural characteristics as woven fabrics do. Rather, they are formed by the deposition of fibres on a conveyor belt and are bonded together in a network using different technologies, such as TAB and TC. Thus, they have a unique type of microstructure, making their characterisation challenging. Fibre crimp is one of the main parameters that geometrically characterises RFNs; a novel, parametric algorithm was developed to compute the fibre crimp in this study. Generally, fibre crimp is calculated [21,22] as the ratio between the total fibre length (L_curve_) and the distance between its both endpoints (L_Straight_). In this research, the equation below was applied to a fibre segment between its endpoints.
Fibre Crimp = L_Curve_/L_Straight_(1)

The developed algorithm was used to compute fibre crimp for nonwovens, which were manufactured with the various parameters presented in Table 1. This algorithm is flexible and parametric for computing the fibre crimp for any type of fibrous networks with various fibre shapes, diameter variations, number of layers, fabric finishing, etc. Most of the existing algorithms for the fibre crimp employ 2D images. As such images are not effective for fibres lying underneath the top layer, these fibres are not detected and counted. High numbers of curly fibres, including 3D twists, can also lead to inaccurate and misleading results as 2D images are not sufficient to present an overall view of the curliness of fibres inside a fabric. This inaccuracy grows with an increase in the density and the number of layers. However, the algorithm presented in this paper uses 3D images to overcome the above-mentioned limitations. The model described here was based on X-ray µCT models, reconstructed from 2D slices. Hence, it is flexible and can detect all 3D, geometric characteristics of a fibre, including 3D, twisted paths in a random structure. The introduced models were intended for nonwovens manufactured with two different manufacturing methods, i.e., TAB and TC. Both of these models contain random bond points. TC webs were bonded using a hot, thermal calendar to create bond points in specified locations on the fabric. However, fibres which were outside of these bonded regions may also be bonded due to the heat of the metal calendar. The algorithm detects all these bond points before proceeding with the fibre-tracking stage.

The X-ray µCT models obtained for the test specimens and parameters were imported into Matlab 2020a software for processing, as shown in Figure 2. The model was filtered using a 3D Gaussian filter to remove salt-and-pepper noise [10]. Following the steps of the process, the de-noised image was binarized, and all the voxels relating to the fibrous structure were denoted as “1”, while the remaining voxels were treated as voids. VGSTUDIO MAX 3.1 Software, integrated with the X-ray µCT system, was used to find an optimum binarization value for the scanned RFNs, while a 3D, volumetric image-processing toolbox in Matlab 2020a software was used throughout the working process. Medial axes of the fibrous structures were acquired using the skeletonization process to track the fibre paths in the binarized images. The skeleton of the fibrous structure was acquired using the “*bwskel*” function of the toolbox. Skeletonization is a thinning process which removes boundary pixels until all the objects in the image become 1 voxel wide centrelines of a fibre. The variability of the cross-section along a fibre segment is negligible and is not considered in this research as the skeletonization process generates a single voxel line (medial axis) for both thick and thin fibres. Later, a pruning operation was implemented to remove the unwanted short particles/branches, as is shown in Figure 3, which were not larger than 20 voxels. Hence, red-colour lines represent fibres, while the yellow-colour lines represent spurious paths. The pruning algorithm is flexible enough to adjust this threshold value, which controls the minimum branch length. To increase the accuracy of the results, the entire computation process was performed for fibre segments. All the branch points were detected and their 3D coordinates were acquired.

Morphological functions named “*endpoints*” and “*branching points*” were used to define the endpoints and the branching points, respectively, of the fibrous structure in 3D. Functions called “*regionprops3*” of the toolbox were used to acquire the coordinates of the intersection and endpoints. To compute the fibre crimp, fibre segments were tracked individually, accounting for their endpoints.

The schematic diagram in (Figure 4) shows the definition of branch points and fibre segments. The curve length of a fibre, L_Curve_, was calculated based on the number of voxels used to generate that segment. Simultaneously, the algorithm detected the free endpoints of fibres and recorded their 3D coordinates. An additional algorithm was created to generate an artificial voxel line between two free endpoints. The number of voxels, necessary to create a pseudo voxel line between these free endpoints of a fibre segment, was stored as L_Straight_. Once all the data were stored to compute the fibre crimp, the virtually created voxel line was removed from the fibrous domain since it could lead to fibres overlapping and thereby affect the calculations. The same process was continued for the next fibre segment until all fibre segments were tracked. Finally, the stored values of “L_curve_” and “L_straight_” were used to compute the crimp degree of each fibre segment.

## 4. Results and Discussion

This section includes the results for the fibre crimp of artificially developed, geometrical models described in the previous section and the X-ray µCT models of the nonwoven webs. Initially, geometrically developed, voxel-based straight lines and semi-circles (Figure 5a,b, respectively) were used to evaluate the developed fibre crimp algorithm. The crimp of the geometrical, voxelated models developed in 3D Matlab were computed using two approaches: analytical and numerical. In the analytical method, the segment length was computed mathematically, and the distance between endpoints was computed using the Pythagorean formula. The crimp was subsequently computed using the above two values. In the numerical method, the segment length was computed by counting the total number of voxels. After detecting the endpoints of the geometric element, a voxel line was created between those points and the number of voxels forming it was computed. Finally, the crimp was calculated as the ratio between those two values. Table 2 presents the analytical and numerical crimp values for the voxelated straight line and two different semi-circles. Apparently, the error level was below 3.0559%. Before applying the algorithm to the six X-ray µCT nonwoven models mentioned in Table 1, it was used for a small RFN with only four fibres. The model was created using CTPro3D software by cropping an X-ray µCT model of 17 gsm crimped fabric (Table 1); the skeletonized model is shown in Figure 6. Results for the three main categories of the fabrics are presented in the form of histograms of the 3D fibre crimp distributions in Figure 7 and Figure 8. An image of each X-ray µCT model is presented alongside the respective histogram. The bin size of all histograms was 0.05; this can be changed in the method according to user preferences. Fibre frequency is shown on the Y axis using a logarithmic scale, and crimp is shown on the *X*-axis. The fibre crimp values for straight fibres are equal to 1; for slightly curved fibres, the values exceed the unity.

The projected crimp values were computed for all three planes (MD-CD, CD-TD and MD-TD) for each nonwoven fabric using the relevant coordinates of repetitive voxels. The projected crimp was computed as follows.
Crimp _(CD-TD)_ = Projected length in voxels in CD-TD/Distance in CD-TD(2)
Crimp _(MD-TD)_ = Projected length in voxels in MD-TD/Distance in MD-TD(3)
Crimp _(MD-CD)_ = Projected length in voxels in MD-CD/Distance in MD-CD(4)

The 2D and 3D fibre crimp were computed for the small-size RFN for all three planes. The results were statistically analysed in terms of the mean and the standard deviation, based on the data for separate fibres. The average value for CD-TD was higher than the average value for other 2D projections, with the lowest value exhibited by MD-CD. The average 3D crimped value was close to the MD-CD projected crimp (Table 3).

The calculated mean values of the 3D fibre crimp for all six models were nearly the same and were close to unity (Figure 9a), meaning that the fibre segments in the studied RFNs were almost straight. The fibres in the first bin (1–1.05) of the histograms were considered nearly straight [23,24,25]. As expected, the fraction of the straight fibres in the crimped nonwovens was less than the fraction in the non-crimped structures (Figure 9b). Apparently, the fraction of straight fibres decreased as the density in both types of fibrous structures increased. In crimped nonwovens, the fibres were originally crimped in their manufacturing technique, followed by TAB and TC. The gradual increase in fibre crimp and the decline of the fraction of nearly straight fibres in both the crimped and non-crimped networks as the density increased can be explained by the effect of out-of-plane fibres [13].

For the given sets of fibres, there were no perfect straight fibre segments, i.e., with a fibre crimp equal to 1, even for non-crimped fabrics; the minimum crimped values for all types of RFNs were slightly higher than 1 (Figure 10a). The maximum values of crimp in each fabric grew with the fabric density (Figure 10b); the highest recorded value was 1.65 for the 90 gsm crimped, nonwoven fabric.

For all six X-ray µCT nonwovens, 2D and 3D crimps were computed and the results were statistically analysed. The average values for both crimped and non-crimped fabrics are presented in Table 4 (detailed results are given in the Appendix A). All the MD-TD, CD-TD, and MD-CD crimps are presented as histograms (Figure 11 and Figure 12) after removing rare events (bins smaller than 0.5%). The values for the CD-TD projection were the highest, while the lowest values were found for the MD-CD projection; similar results were observed for the small-size X-ray µCT of the nonwoven model.

The mean values of 3D fibre crimp were nearly equal to those in the projection of the MD-CD planes. Due to the nature of the manufacturing process, fibres were laid down on a conveyor belt from a hopper and were mainly distributed in the MD-CD plane. Therefore, the 3D geometrical behaviour of materials such as nonwovens is mostly defined by the geometry in the MD-CD plane. The standard deviations and mean values of 3D fibre crimp and MD-CD projection for crimped nonwovens was always higher than those of the non-crimped fabrics (Table 4).

The projected fibre crimp for all types of nonwoven fabrics had the highest values in the CD-TD plane and lowest values in the MD-CD plane (Table 3 and Table 4). When fibres are oriented out of plane (in direction of the TD axis), they have more curls and twists, increasing their crimp. Additionally, fibre segments create higher inclination angles along TD axis when they move out of plane [13]. For sufficiently high conveyor belts, fibres are orientated along the MD axis in the manufacturing process as the conveyor belt moves along the MD [26]. As a result, the fibre crimp in the MD-CD plane was the lowest; there was no contribution from the TD axis. Additionally, when fibres are deposited onto the conveyor belt layer-wise, the friction between the conveyor belt and the fibrous network (MD-CD plane) stretches the fibre segments in-plane. As the conveyor belt moves along the MD, fibres are more likely to be stretched along the MD and partially on the CD as it is an in-plane axis.

The obtained statistical results (standard deviation and mean values) show that the projected crimp increased when the TD axis was involved due to the general manufacturing process followed by TAB and TC (Table 4). As fibres moved out of plane, the crimp exceeded the MD-CD values. The highest crimp values were observed on the CD-TD projection as it involved the out-of-plane direction. Additionally, due to the manufacturing process, the fibre orientation along the CD was lower than along the MD. Similarly, statistical results (standard deviation and mean values) showed that the high crimp values were observed on the CD-TD projection and the lowest were observed on the MD-CD projection for both crimped and non-crimped fabrics (Table 4). The effects of heat and pressure along the TD in TAB and TC fabrics led to additional crimp in the TD. As a result, hills and valleys were formed across the TD, which led to a crimp contribution along the TD. Therefore, the crimp in the CD-TD projection was the highest and the crimp in the MD-CD plane was the lowest. However, the 3D crimp values showed the real crimp of an RFN, and fibre segments between intersections were almost straight. The projected crimps overestimated the real crimps in RFNs such as nonwovens.

The projected crimp distribution is presented in the diagram in Figure 13. For instance, TD was considered “High” as it tended to generate high crimp due to its out-of-plane direction. CD was considered “Medium” and MD was scored as ”Low”, as explained above. Therefore, the values for the CD-TD crimp were “High-Medium” and scored the highest. The values for MD-TD were “High-Low”. The lowest score was given to the MD-CD projected crimp (“Medium-Low”).

The obtained fibre crimp results are important for the evaluation of the mechanical properties of RFNs such as nonwovens because they are significantly affected by fibre crimp [18]. Additionally, fibre crimp has a significant impact on the stiffness of a fibrous structure: a high fibre crimp is related to a lower stiffness [27]. The results for the projected fibre crimp results computed in this study are useful to study the in-plane and out-of-plane stiffness of RFNs. Products for applications such as medicine and hygiene, for which fibre stiffness is important and can be hard to predict, can be further improved using the findings in this study. For instance, medical dressings that require a specific stiffness in certain directions can be easily evaluated using the developed technique.

The assessment of fibre crimp is important as it affects the fabric’s pore-size distribution and increases the number of bonds per fibre. Hence, it has a high impact on isotropic and anisotropic structures. For instance, when fibres are more twisted in a 3D space (deviating from a straight line), increasing the waviness, both the surface area and the number of fibre segments per volume grow compared to a fibrous structure with straight fibres. Evidently, this reduces the porosity of a fibrous structure [22], making a significant impact on its permeability. The results of this study can be used to evaluate and analyse the impact of fibre crimp on the permeability characteristics of RFNs. The developed algorithm also allows for the understanding of the overall fibre crimp of a fabric. Generally, fabrics with high numbers of curled fibres break down in a wide range of strain [18]. Thus, fibre crimp results obtained with the developed algorithm can be used to evaluate the tensile behaviour of RFNs, and most of the tensile properties of nonwovens were tested in the MD-CD plane experimentally and numerically [26,28,29].

## 5. Conclusions

In this study, the computation of fibre crimp was performed using 3D microscopical images of RFNs for the first time. The suggested novel parametric algorithm was tested using the first virtually developed known geometrical models. It was also able to successfully compute the levels of fibre crimp for a variety of random, fibrous structures with different densities. This algorithm was successfully tested for both crimped and non-crimped materials. The employed fibrous models had complex bonding in TAB and TC webs, which led to random distributions of bond points. This method could be applied to compute the fibre crimp of random structures with fibres of varying diameters made with different materials such as metal/polymer fibres or wood and biological systems such as collagens. To improve the accuracy of the results, the fibre crimp was computed for fibre segments distributed between bond points. The crimped value of a fibre can be changed by the manufacturing process. For the given set of RNFs, it was observed that the fibre crimp increased in denser structures. On the other hand, there was a gradual decrease in the fraction of straight fibres in non-crimped nonwovens when the density increased. At a higher density, fibres have more freedom to occupy the 3D space in an RFN [13]. For both crimped and non-crimped networks, the CD-TD crimp was highest, and the MD-CD crimp was the lowest. The 3D fibre crimp was nearly equal to the MD-CD crimp, with almost-straight fibre segments between intersections with a slight curliness. This novel algorithm could influence multiple fields such as CFD to optimize the permeability of random fibrous structures based on fibre crimp, using FEA (finite element analysis) to verify or validate the mechanical behaviour of fibrous structures.

## Figures and Tables

**Figure 1 polymers-15-01050-f001:**
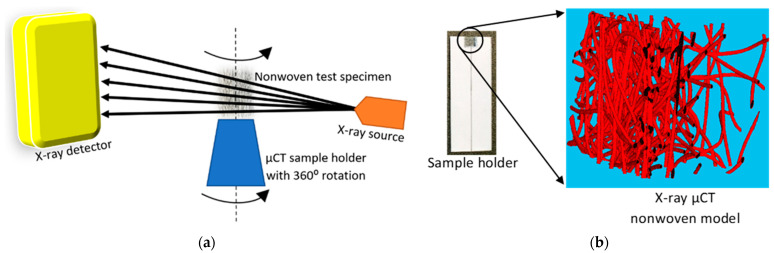
X-ray µ-CT system: (**a**) working procedure of X-ray µCT and (**b**) test specimen.

**Figure 2 polymers-15-01050-f002:**
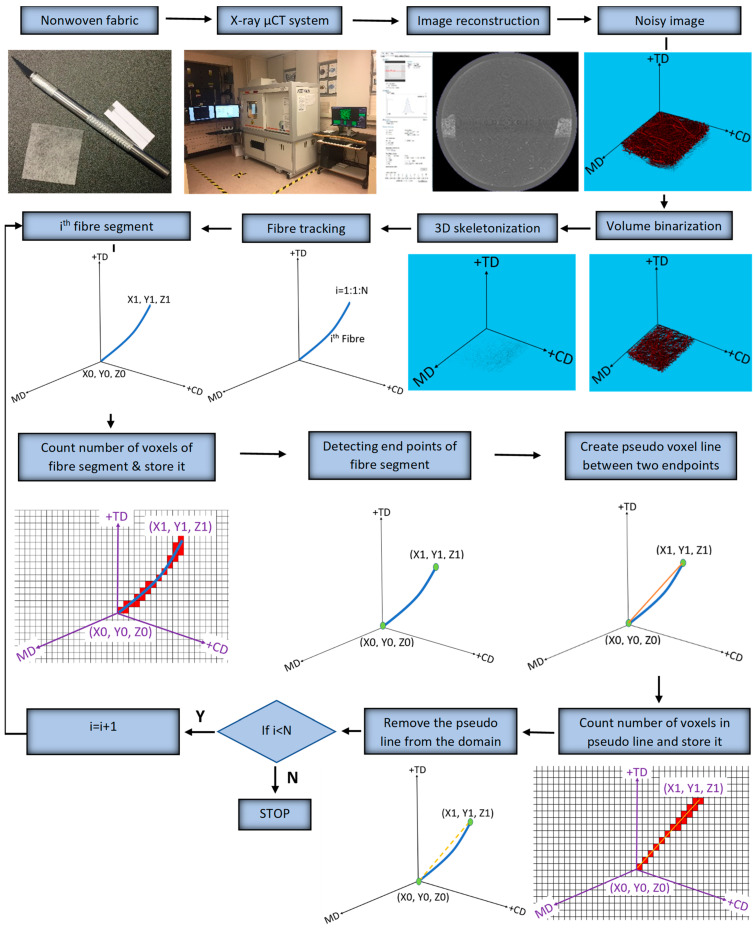
Process flow chart: i represents the fibre number, and i = 1,2, 3,…N, where n indicates the total number of fibre segments in the nonwoven model.

**Figure 3 polymers-15-01050-f003:**
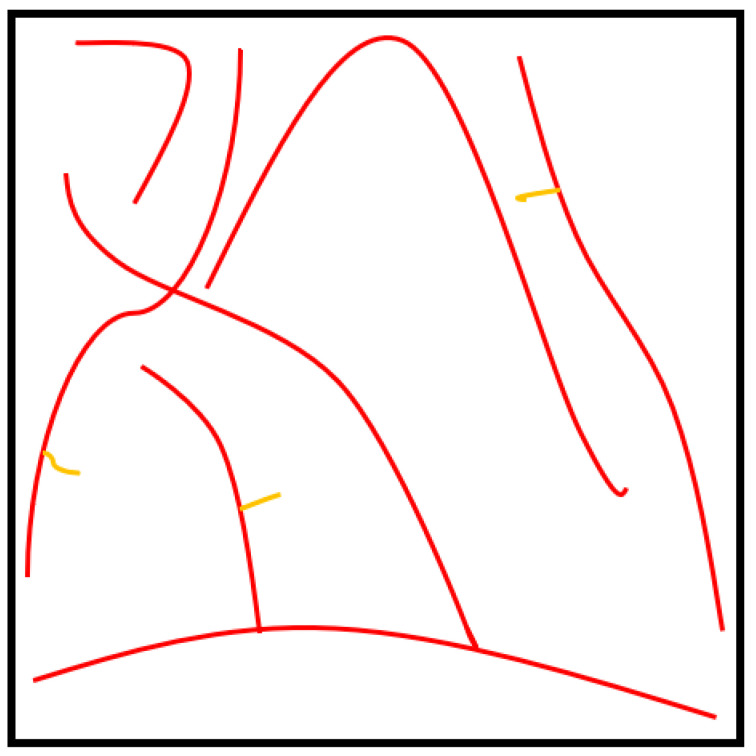
Pruning operation.

**Figure 4 polymers-15-01050-f004:**
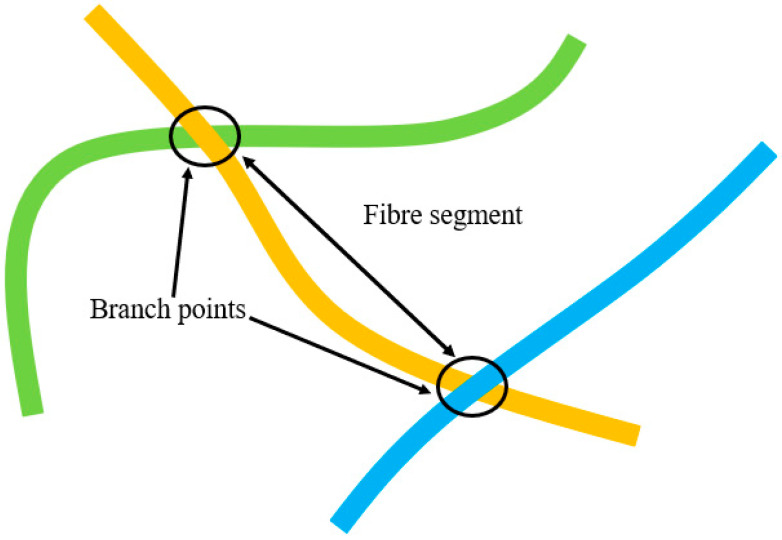
Branch points and fibre segments.

**Figure 5 polymers-15-01050-f005:**
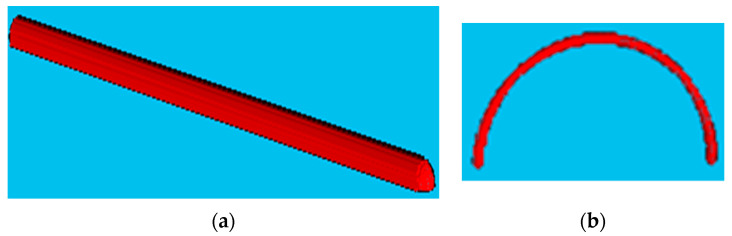
(**a**) Voxelated line and (**b**) voxelated semi-circle.

**Figure 6 polymers-15-01050-f006:**
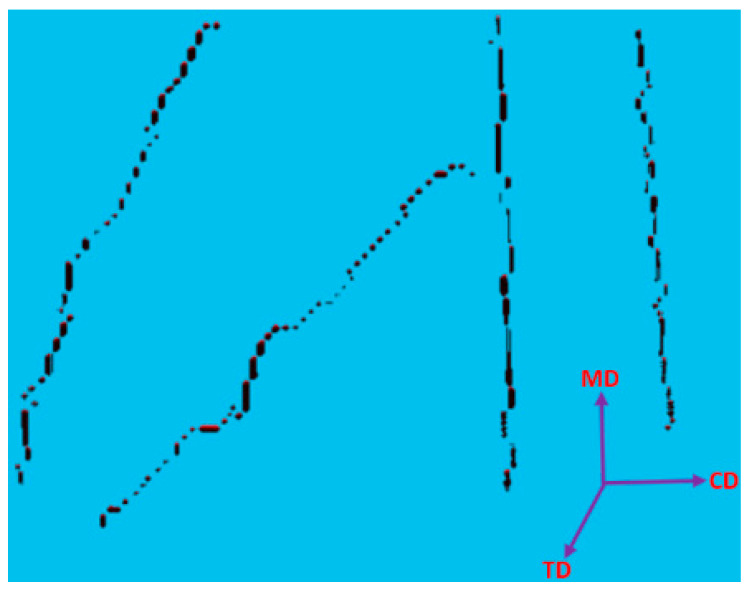
Skeletonized, small-size X-ray µCT nonwoven model with four fibres.

**Figure 7 polymers-15-01050-f007:**
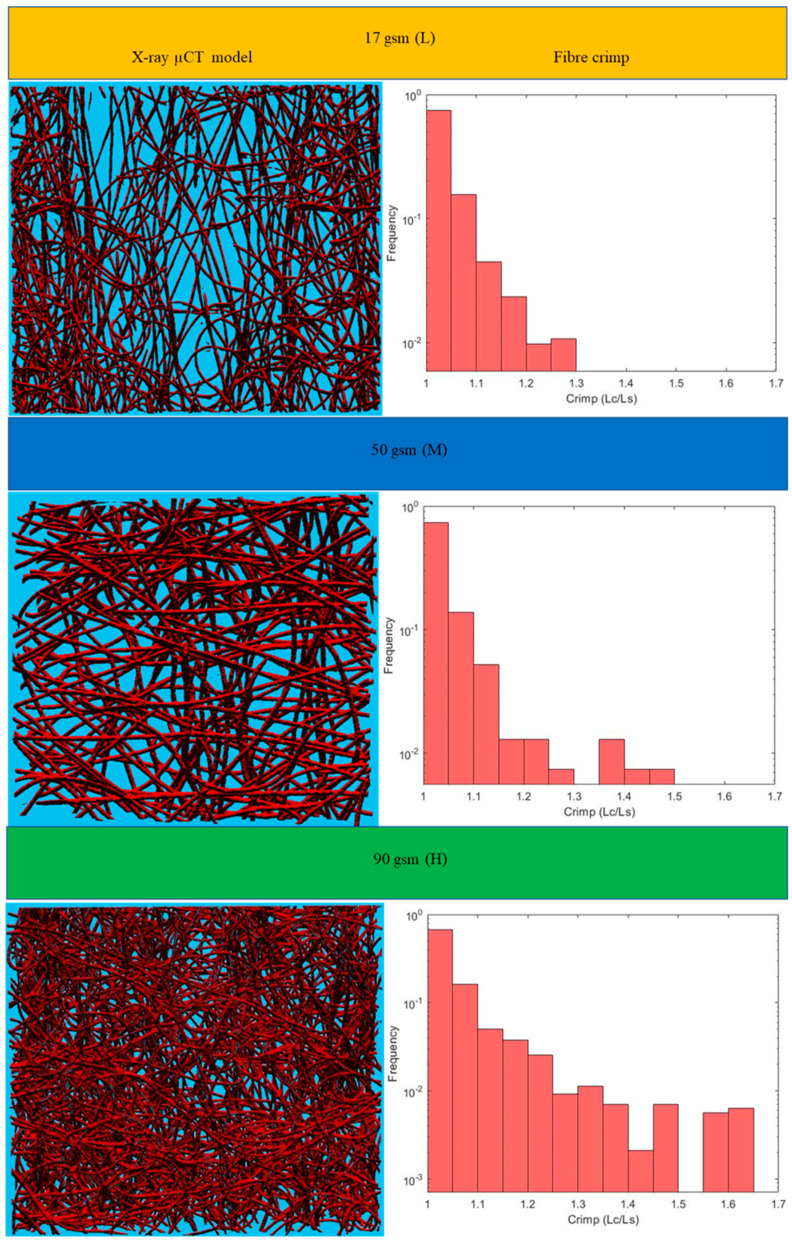
Distributions of 3D fibre crimp for fabrics with crimped fibres.

**Figure 8 polymers-15-01050-f008:**
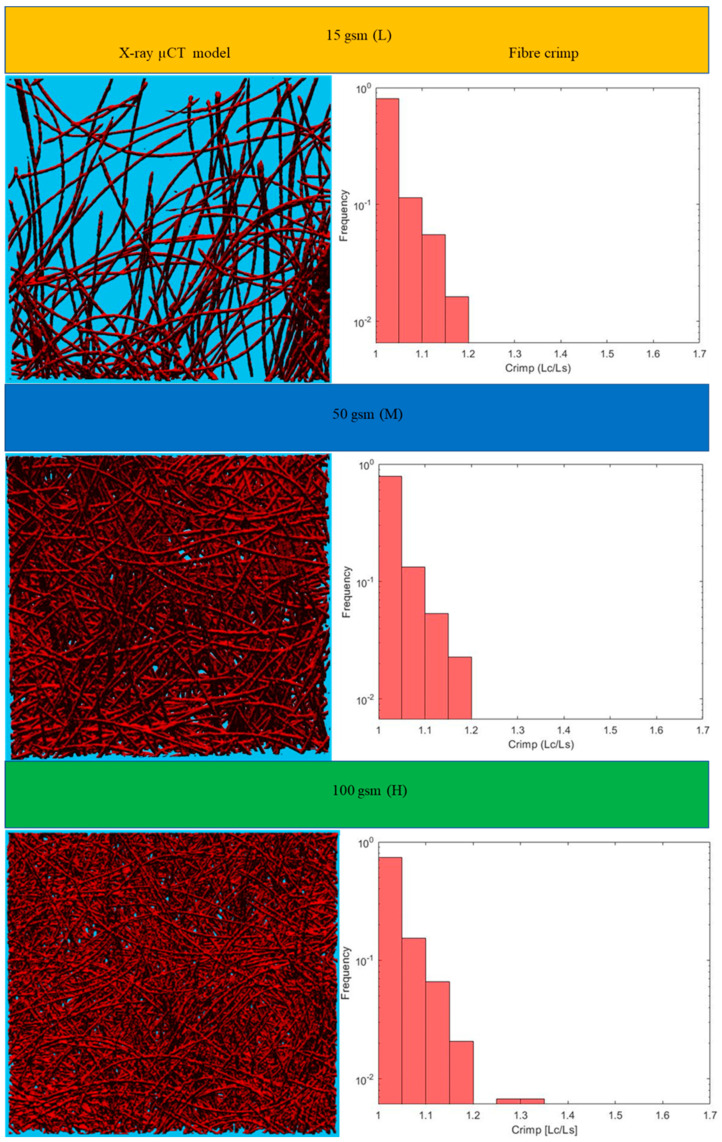
Distributions of 3D fibre crimp for fabrics with non-crimped fibres.

**Figure 9 polymers-15-01050-f009:**
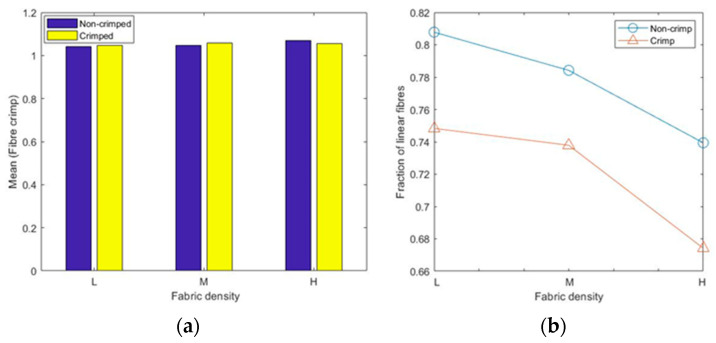
Fibre crimp results: (**a**) mean crimp and (**b**) fraction of linear fibres.

**Figure 10 polymers-15-01050-f010:**
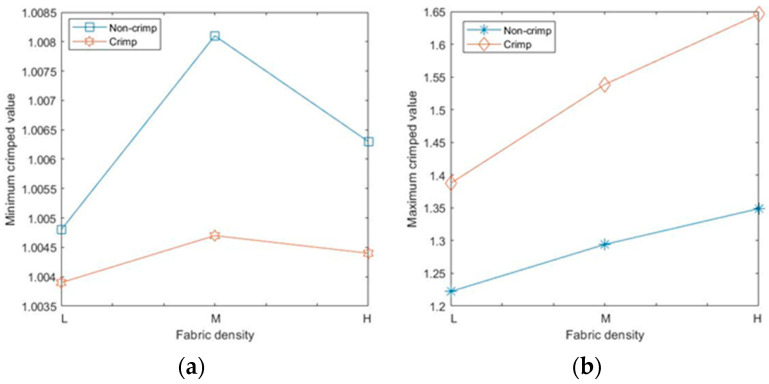
Fibre crimp results: (**a**) minimum values and (**b**) maximum values.

**Figure 11 polymers-15-01050-f011:**
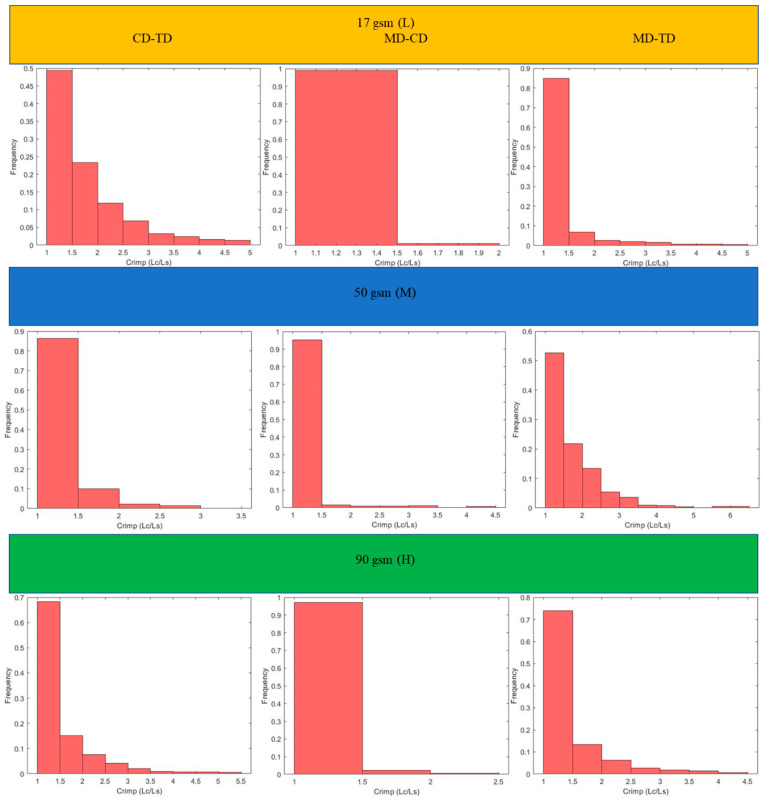
Two-dimensional crimp of crimped fabric.

**Figure 12 polymers-15-01050-f012:**
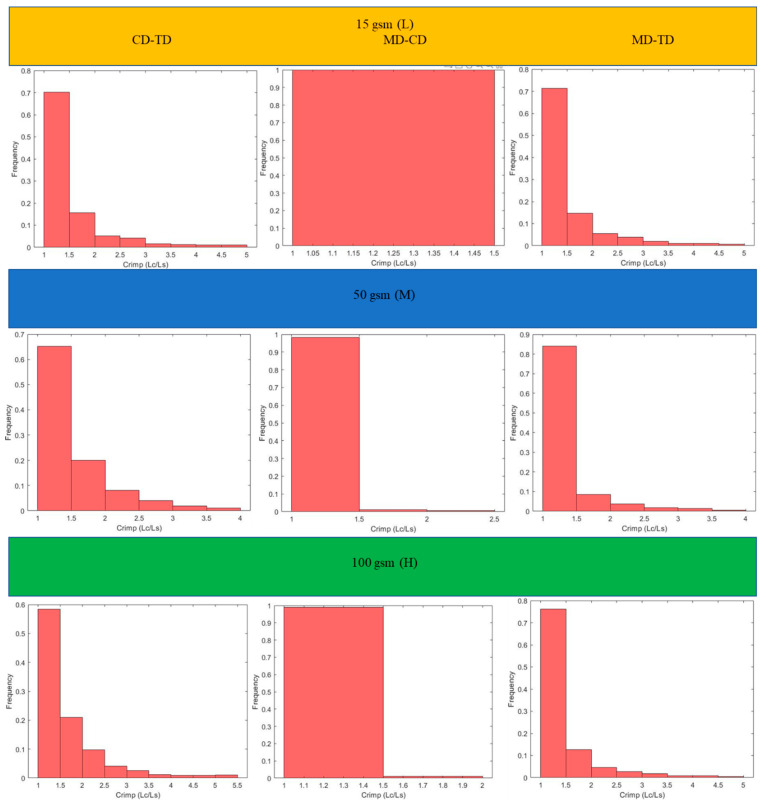
Two-dimensional crimp of non-crimped fabrics.

**Figure 13 polymers-15-01050-f013:**
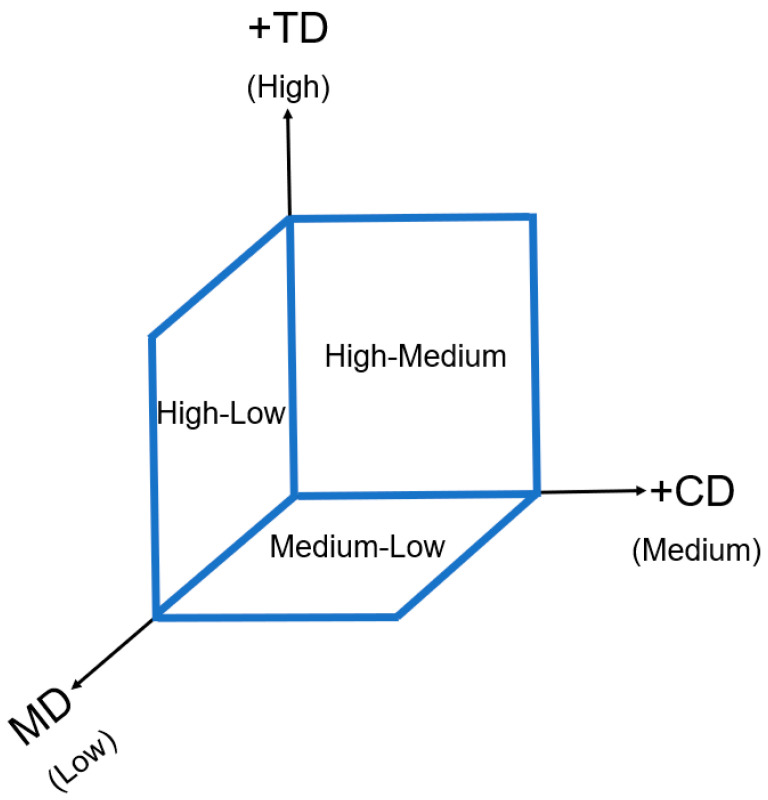
Projected fibre crimp.

**Table 1 polymers-15-01050-t001:** Manufacturing parameters of nonwoven fabrics used.

Category	Density (gsm)	Type	Material	Diameter * (µm)	Number of Layers	Crimped (Y/N)	Finishing
L	17	TAB	PP/PE	22	2	Y	Fine
	15	TC	PP	15	3	N	Fine
M	50	TAB	PP/PE	37	1	Y	Coarse
	50	TC	PP	19	3	N	Fine
H	90	TAB	PP/PE	41	2	Y	Fine
	100	TC	PP	20	3	N	Fine

* Measured with SEM.

**Table 2 polymers-15-01050-t002:** Analytical and numerical crimp values of voxelated models.

Model Type	Analytical Value	Numerical Value	Error (%)
A straight line	1.0000	1.0000	0
Semi-circle (Radius = 25)	1.5707	1.5397	1.9736
Semi-circle (Radius = 35)	1.5707	1.5227	3.0559

**Table 3 polymers-15-01050-t003:** Statistical data of small-size RFN.

Projection	Parameter	Value
CD-TD	Mean	1.7799
	Standard deviation	0.6181
MD-TD	Mean	1.1425
	Standard deviation	0.1143
MD-CD	Mean	1.1117
	Standard deviation	0.1110
3D crimp	Mean	1.0953
	Standard deviation	0.1107

**Table 4 polymers-15-01050-t004:** Average statistical values of X-ray µCT models.

Projection	Parameter	Crimped	Non-Crimped
CD-TD	Mean	1.6805	1.5505
	Standard deviation	0.7681	0.6748
MD-TD	Mean	1.3451	1.3900
	Standard deviation	0.5116	0.5789
MD-CD	Mean	1.1311	1.0952
	Standard deviation	0.1822	0.0795
3D crimp	Mean	1.0583	1.0465
	Standard deviation	0.0766	0.0385

## Data Availability

Where data is unavailable due to privacy.

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
