# Peer review of "Assessing Crimp of Fibres in Random Networks with 3D Imaging"

_polymers, 2023, doi:10.3390/polym15041050_

Round 1

Reviewer 1 Report

The report presents an analysis of the morphology of fibres within random fibrous networks by using a newly developed algorithm based on the data of micro-CT scanning. The methodology developed in the work is novel and the obtained results are rational. However, there are still some issues to be addressed before proceeding to a publication.

1.       Page 3, line 123, it says “the test specimen with a 3mm x 3mm window size”. But the specimen (Figure 1b) does not look like a square.

2.       Page 3, as the 3D models are reconstructed by 2D images, it is suggested to specify the thickness of the specimens.

3.       Page 4, line 160, what is the statistical/average length of the fibres ? And please comment if Equation 1 could represent the real morphology of fibres, as a long fibre could consist of a series of curves.

4.       Page 5, line 193, it says “orange colour”, but it looks like yellow in Figure 3.

5.       Page 6, Figure 2, the images for pruning and 3D skeletonization are not informative, please improve the quality.

6.       Page 9, Table 3, please comment why 2D projects are worthy of discussions as 3D models could describe the fibres more realistically.

7.       Page 13, Figure 9a, it is suggested to use 0.9-1.1 range for the vertical axis to highlight the difference.

8.       Page 16, it is not suggested to use a whole section for “potential applications” in an academic paper, the contents could be merged in sections of introduction and conclusion.

Reviewer 2 Report

The paper is well written, and it can be published with the present form, except some correction, if authors are interest to do so,

For example, Fig 1b, nonwoven model, means what kind of nonwoven, spun bonded or, please specify. The model explained in the Fig 2 is generalised model or specific fibers? since cross sections for the cotton and polyester are different. 

Reviewer 3 Report

Dear authors,

Thanks for the manuscript, but I have some comments, questions and modifications that need to be done. 

1- you wrote that you study crimping, but as I saw in the manuscript, your work is on the curvature and convolution of the fibre. 

The fibre crimp is the waviness of fibre, so if you want to speak about crimp, you need to have a wave, and if you have a wave need to be described; if not, you need to use another word than crimp. 

In figure 5, you presented fibre as a semi-circle.

2- you didn't mention the production method of the non-woven web, and you mentioned only the bonding process. In lines 264, 290, 300, 309, and 311, you wrote about the manufacturing Technique/process, which we don't have any information about it in the manuscript.

3- Fiber length is missing, and all your calculation based on it. 

4- The description of the cross-section of the fibre is missing, and it is an important parameter for your case, and you had mentioned the importance of the crosse section in the literature. 

5- we found the term additive manufacturing in the abstract and the conclusion, and we don't find it in the mean text; it needs to be clarified in the main text. 

6- In lines 294 and 308, "statistical results," is it the standard deviation, the frequency or crimps?

7- For standard deviation, you have a huge difference; some results are very high, and you need to have more discussion about and give the limitation for your method.

8- We have the term linear fibres. figure 9, we didn't find it in the text, is the straight fibres? and you also wrote the term line in figure 5?

9- The effect you obtained on the fibre is directly related to the slenderness ratio (stiffness) slenderness ratio = Fibre length/Fibre diameter, which explains clearly your results. And this completed missing in your manuscript.

10- For the structure, it is clear to use a material and methods, with clear descriptions, then your structure, where you have used "System Set-up".

Best Regards

Round 2
